# Machine Learning in Unmanned Systems for Chemical Synthesis

**DOI:** 10.3390/molecules28052232

**Published:** 2023-02-27

**Authors:** Guoqiang Wang, Xuefei Wu, Bo Xin, Xu Gu, Gaobo Wang, Yong Zhang, Jiabao Zhao, Xu Cheng, Chunlin Chen, Jing Ma

**Affiliations:** 1Key Laboratory of Mesoscopic Chemistry of MOE, School of Chemistry and Chemical Engineering, Nanjing University, Nanjing 210023, China; 2Department of Control Science and Intelligent Engineering, School of Management and Engineering, Nanjing University, Nanjing 210093, China; 3Jiangsu Key Laboratory of Advanced Organic Materials, School of Chemistry and Chemical Engineering, Nanjing University, Nanjing 210023, China

**Keywords:** automatic chemical systems, machine learning, coordinated multi-robot systems, virtual screening

## Abstract

Chemical synthesis is state-of-the-art, and, therefore, it is generally based on chemical intuition or experience of researchers. The upgraded paradigm that incorporates automation technology and machine learning (ML) algorithms has recently been merged into almost every subdiscipline of chemical science, from material discovery to catalyst/reaction design to synthetic route planning, which often takes the form of unmanned systems. The ML algorithms and their application scenarios in unmanned systems for chemical synthesis were presented. The prospects for strengthening the connection between reaction pathway exploration and the existing automatic reaction platform and solutions for improving autonomation through information extraction, robots, computer vision, and intelligent scheduling were proposed.

## 1. Introduction

As the core of chemical science, synthetic chemistry refers to creation of new molecular structures (or new substances) with specific functions via one or a series of physical or chemical operations in combination with certain characterization and analysis techniques. It is closely related to energy, the environment, and human health, etc. However, creation of a target molecule often requires professional researchers and highly intensive experimental attempts. This process includes: (1) researchers generate a synthesis plan based on personal chemical intuition from mechanistic understanding or experience; (2) executing synthetic experiments; (3) reaction outcome analysis (e.g., characterization of the structure and properties of the product, yield, selectivity, etc.) (Figure 1a). In this process, high-dimensional parameter spaces (e.g., reactant, catalyst, solvent, etc.) need to be considered, and, therefore, it requires experimenters to explore several condition combinations. This research paradigm not only leads to high labor and material costs but also makes researchers physically tired because of long working-time and repeated work, which might further cause reproducibility and experimental safety problems. It is our dream to build a smart unmanned system that consists of virtual screening, task assignment, automatic experimentation (synthesis and characterization), data feedback, and self-optimization. Such a closed-loop system is driven by machine learning (Figure 1b) through a cyber-physical system (CPS) platform.

To liberate experimenters from those routine tasks, many endeavors have been made to promote automation and intelligence of automatic chemical systems. The first automated chemical experiment system appeared around the 1970s (Figure 2). Computers were introduced to control delivery of chemicals; e.g., Deming et al. developed a computer-program-controlled automatic synthesis system that dispenses chemicals through a controlling injection pump and realizes automatic synthesis and characterization of specific products in combination with a UV–vis spectrophotometer [1]. Later, Legard and Foucard developed a general automatic toolkit, Logilap, which can select and match different components according to the requirements of the experiment [2]. With development of science and technology, the automation degree of synthetic laboratories has increased rapidly, which can deal with many different types of chemicals and reactions. Frisbee et.al designed an automatic experimental system with multiple reactors and mechanical arms that carried out experiments in parallel. The system could not only perform synthetic operations, such as quenching, extraction, and filtration, but was also equipped with an automatic cleaning machine [3], which further improves the degree of autonomation.

With rapid development of the automation laboratory, development of autonomous discovery systems with the ability to think independently is arousing widespread interest [4]. At this time, machine learning has stepped onto the stage in the field of chemistry. Here, applications of machine learning in the field of unmanned chemical systems are introduced, with a focus on the prospect of smart reaction design and synthesis systems.

## 2. Applications of Machine Learning to Unmanned Chemical Systems

### 2.1. Categories of Machine Learning Models

Through building mathematical models, machine learning (ML) can learn related tasks from data. Recently, ML has been widely used in medical, chemical, computer vision, and many other fields [5,6,7]. Machine learning is a branch of artificial intelligence that learns by the training set in a certain way. With the amount of training increasing, the performance of ML is gradually optimized, which predicts output of related problems. The model, the strategy, and the algorithm are described as the three elements of ML. The model is used to analyze the problems that need to be solved, which is mapping the relationship between input space and output space, such as non-linear models, linear models, and regression models, etc. The strategy is to select the learning criteria for the optimal model; e.g., regression can use mean square error (MSE) as the learning strategy. Here, the loss function is introduced, which measures the error between the predicted and true results. The algorithm is a specific calculation method for the model to learn, which solves optimization problems. The common optimization algorithms are gradient descent, Newton’s, and quasi-Newton methods, etc. Then, to adjust the model parameters and prevent the model from overfitting, the optimal hyperparameters of the model are selected by using the validation set, and the generalization ability of the model is evaluated by using the test set.

In general, the field of machine learning can be divided into three categories: supervised learning, unsupervised learning, and reinforcement learning (Table 1).

Supervised learning [8], which requires training with labeled input and output data, aims to learn or approximate a mapping function that can best describe the association patterns and relationships between input features and output value. Supervised learning methods can be further divided into two subdomains: classification if the labeled outputs are discrete variables and regression, if the labeled outputs are continuous variables. Once this function is well-trained and converged, it can be utilized to predict the output value for unseen input data. In supervised learning, some representative algorithms are support vector machine [9], naive Bayes [10], hidden Markov model [11], and Bayesian networks [12].

In contrast, unsupervised learning aims to infer the inherent structure of training data without any available labels [13]. According to how the data are processed and analyzed, unsupervised learning methods can be further divided into two categories: clustering, which classifies input data with similar features or properties into the same group; and dimensionality reduction, which reduces the dimension of input data into a relatively smaller set of features with minimum loss of information and performance. As for unsupervised learning, the representative methods are K-means and X-means [14], Gaussian mixture model, and Dirichlet process mixture model [15].

Reinforcement learning [16] aims to allow an autonomous active agent to learn its optimal behavior policy to complete a certain task by maximizing cumulative reward in a trial-and-error manner while interacting with an initially unknown environment. Classical algorithms, such as dynamic programming [17], Monte Carlo methods [18], and temporal-difference learning [19], have witnessed great success in Markov decision processes (MDPs) with discrete state and action space. Recently, with development and partnership of deep learning, deep reinforcement learning (DRL) is capable of addressing high-dimensional problems ranging from video games [20,21] and board games [22] to robotic control tasks [23,24].

**Table 1 molecules-28-02232-t001:** Categories of machine learning models and related applications in chemical synthesis.

Categories	Methods	Applications	Ref.
Supervised learning	Support vector machine, Multivariate linear regression (MLR), Decision trees, etc.	Reactivity prediction, Chemical reaction classification, Autonomous research system, etc.	[25,26,27,28,29,30,31,32,33,34,35]
Unsupervised learning	K-means and X-means, Gaussian mixture model, Dirichlet process mixture model	Information extraction, Molecular Simulation	[36,37]
Reinforcement learning	Temporal difference, Q-learning, Deterministic policy gradient, etc.	Robotic control, Synthetic route plan, etc.	[23,24,38,39,40,41,42,43,44,45,46,47,48,49,50,51,52,53,54,55,56,57,58,59,60,61,62,63,64,65,66,67,68]
Advanced learning	Deep learning, etc.	Natural language processing, Property prediction, Catalyst design, etc.	[69,70,71]

In addition, numerous machine learning algorithms have been proposed in the past few decades. Here, we briefly introduce some advanced learning methods that may be either promising or often used in unmanned chemical robotic systems (Table 1), which have been widely used in ML communities through combination with the aforementioned three typical ML methods. For example, deep learning has grown to be one of the most popular research trends in the field of machine learning. Due to its well-performed deep architectures, deep learning methods have managed to realize learning hierarchical representations end-to-end and achieve state-of-the-art performance of capturing the statistical patterns of input data [72]. Hence, they have attracted extensive attention from both the academic community and industrial applications, especially in the field of computer vision and natural language processing [73,74,75]. In the field of chemistry, a multilevel attention graph convolution neural network, called DeepMoleNet, has recently been applied to predict energies of hydroxyapatite nanoparticles (HANPs), quantum chemistry properties of organic molecules, and reaction energy of nitrogen reduction reaction process in metal-zeolites [69,70,71].

Representation learning aims to extract and organize discriminative information from high-dimensional input features [76]. To be specific, representation learning aims at obtaining a reasonably sized learning representation that captures several high-dimensional input features to significantly improve computational efficiency and statistical efficiency [77].

Transfer learning is proposed to extract knowledge from source tasks that have been met before and apply the learned knowledge to an unseen target task, enabling domains, tasks, and data distribution to be different [78,79]. Recently, transfer learning has been successfully applied to many real-world applications, such as constructing informative priors, cross-domain text classification, and large-scale document classification [80,81,82].

Distributed learning aims to scale up learning algorithms to use massive data to learn within a reasonable time by allocating the training process among several workstations [83]. Different from the traditional learning framework that gathers data into a single workstation for central learning, distributed learning allows the learning process to be carried out in a distributed manner for managing a large amount of data [84].

Active learning attempts to deal with the situation where the labeled data are sparse and hard to obtain in real-world applications, aiming to select the most critical instances to complete labeling and training [85]. It has been verified that active learning can achieve higher classification accuracy using as few labeled data as possible via query strategies than traditional learning methods [85].

### 2.2. Chemical Automation

Automation has emerged as a highly efficient strategy to conduct routine operations in chemistry, and recent advances in robotics and computing capability have greatly facilitated development of chemical automation. In material science, many functional materials or drugs are discovered serendipitously. However, due to the vastness of chemical space (estimated to be larger than 10^60^), it is a great challenge for material discovery [86]. The combination of ML and chemistry would narrow the gap between the infinite chemical space and finite synthetic capacity. The ML methods commonly used in chemistry include multivariate linear regression (MLR), decision trees, random forests (RF), gradient tree boosting consider, k-nearest neighbors and support vector machines (SVM), and deep learning.

Introduction of ML to unmanned chemical systems would construct an autonomous discovery system, as shown in Figure 3 [25]. The general workflow of the autonomous discovery system includes generating hypothesis, testing, and adjusting the hypothesis according to the feedback from automated experiments (Figure 3a). The emergence of autonomous discovery system makes a new leap in the context of chemical automation. The first batch of autonomous discovery systems, such as Adam and Eve, are used in biomedicine, as shown in Figure 3b,c. Adam has sophisticated software and hardware that can also support microbiological experiments, and it can be fully automated to discover new scientific knowledge. Adam used decision trees and random forests to study the function of genes and enzymes [26]. Compared to Adam, Eve focuses on drug screening, and it is designed to be flexible to perform different bio-activity analysis [27].

For chemical synthesis, Steiner et al. developed a system called the Chemputer, which consists of a reaction flask, a jacketed filtration setup capable of being heated or cooled, an automated liquid–liquid separation module, and a solvent evaporation module. This equipment is capable of accomplishing four key stages of a synthetic process: reaction, processing, separation, and purification [4]. To further improve the degree of automation, Fitzpatrick et al. designed a LeyLab system that can complete automatic synthesis and autonomously optimize the reaction parameters [28]. To speed up the efficiency of design, Nikolaev et al. built an autonomous research system (ARES) that utilizes autonomous robotics, artificial intelligence, data sciences, and high-throughput and in situ techniques, and ARES uses the random forest algorithm [29]. Further, Granda et al. proposed an organic synthesis robot with machine learning algorithm SVM to perform chemical reactions, analyses, and predict the reactivity of possible reagent combinations. This approach allows the platform to operate experiments without prior knowledge. The predictive power of this system was demonstrated to be able to predict the reactivity of 1000 combinations with greater than 80% accuracy after considering the results of slightly over 10% of the dataset [30].

## 3. Implementation and Applications of Unmanned Chemical Systems

It is a goal to design an intelligent, unmanned automatic chemical reaction machine with high precision and efficiency. Holistic solutions for machine learning algorithms have also been developed to drive the whole system to work autonomously and intelligently. As shown in Figure 4, the designed system is implemented by two cooperative robots, i.e., a wheeled mobile robot with a manipulator arm for material handling and a manipulator of robotic arm on a guided rail for chemical manipulation, a reactive agent station, and a reaction stage with a magnetic stirring reactor. The capacities of this platform were demonstrated by its application in a condensation reaction of 2,4-dinitrophenyldrazine with formaldehyde and automatic catalyst evaluation of a heterogeneous aza Diels–Alder reaction, suggesting the designed machine is applicable to both homogenous and heterogeneous chemical reactions [87]. The goodness of fit in three robotic operation sets (R^2^ = 0.99857, SD = 0.00174) shows that the repeatability of robotic operation on the automation operation is better than manual operation (R^2^ = 0.98667, SD = 0.00443).

We introduce the potential solution from the chemical task allocation to the completion of the experimental task, which is divided into five stages or functionalities, i.e., information extraction, control of robots, computer vision, intelligent scheduling, and computational-based virtual screening.

### 3.1. Information Extraction

To expand the capabilities of unmanned laboratory systems, we try to make the machines automatically execute the synthetic task following the procedures that are described in patents or literature. To achieve this purpose, the relevant information describing synthetic procedures should be extracted and translated into machine-understandable language. Thus, we need to design automated conversion from unstructured experimental procedures into structured ones: a series of reaction steps with associated properties. Extracting information from text is a problem of text-mining in the field of natural language processing (NLP). Traditional rule-based methods may work if the information is described in a fixed format. However, the description of a synthetic procedure for chemical reactions is generally unstandardized, and, thus, it is unlikely to define rules covering every possible way. Therefore, development of machine-learning models learning from data instead of rules would be more robust to noise in text [88].

Keyword extraction is a branch of text mining that is mainly divided into two categories, i.e., unsupervised keyword extraction and supervised keyword extraction. Unsupervised keyword extraction does not require manual labeling. It selects the candidate words from the text and selects the ones with high scores as keywords, and the scoring strategies include keyword extraction based on statistical features, keyword extraction based on the lexicographic model, and keyword extraction based on the topic model. The key of the supervised keyword extraction method is to train the extractor classifier, which first classifies the candidate words extracted in the article and finally labels the candidate words as keywords. To avoid the high labor costs of supervised methods, unsupervised keyword extraction is the main method. Deep learning is a specialized branch of machine learning and one of its major strengths is its potential to extract higher-level features from the raw input data [89]. Further, sequence-to-sequence models are the most common models used in text-mining nowadays, including recurrent neural (RNN) network, long short-term memory (LSTM) network, and models based on transformer [90] architecture, e.g., bidirectional encoder representations [91].

We currently adopt the BERT model to address our issues, which uses a self-attention mechanism to allow access to all words in the text simultaneously for considering contextual information. The BERT model training process includes two stages: pre-training on unlabeled data and additional training on labeled data for a specific application problem [92]. That is, a classification layer can be added to the pre-trained model, and all the parameters are jointly fine-tuned on a downstream task.

Based on this model, we further divide our task into two subtasks: named entity recognition (NER) and relation extraction (RE). First, we extract all the entities, including the reaction-step, chemicals, solvent, reaction time, temperature, etc., by fine-tuning the pre-trained BERT model. Subsequently, we estimate the relations between pairs of these entities, e.g., time or temperature is the parameter of the specific reaction-step and quantity is the parameter of a chemical or solvent. In this way, we can finally obtain all the sequential steps with associated properties and the relations between them. According to these entities and relations, we map the corresponding reaction-steps to the operations that the machines in our unmanned chemical laboratory can perform to build a complete workflow. In this way, machines can automatically execute chemical reactions following the reported procedures, provided that all the necessary conditions and chemicals are available.

### 3.2. Robotic Control for Unmanned Chemical Systems

Control of Robotic Arm. The robotic arm is considered to be the most critical component in carrying out elaborate operations in unmanned chemical systems [93]. Typical operations of robotic arms include grasping chemical reaction vessels or sample vials and manipulating other instruments. These operations have been realized in one of our independently developed unmanned chemical laboratories, as shown in Figure 5.

In the specific scenario of an unmanned chemical laboratory, the robotic arm is required to manipulate the end-effector towards multiple target positions for further operations at minimum cost. To empower the robotic arm with the intelligence of generalizing manipulation abilities to these tasks, one may use reinforcement learning (RL) [16] framework to learn the optimal behavior policy in a trial-and-error manner. In recent years, researchers have made a series of meaningful attempts at application of reinforcement learning and deep learning algorithms in the field of robot arm planning. Amarjyot et al. [38] comprehensively analyze and compare the task performance of reinforcement learning algorithms in robot arm grasping. Finn Chelsea et al. [39] directly realize the work from image input to targeted capture using multiple long short-term memory networks. Song et al. [40] realize lower training burden and more efficient planning path by ingeniously constructing value function. Devin et al. [41] use the attention mechanism to realize the trajectory prediction of the robot arm to the external moving objects, thus improving the obstacle avoidance ability in the moving scene. Tuomas et al. [42] use the soft Q-learning algorithm to complete the control of a series of complex manipulations of the manipulator, such as building a log tower, and proved the outstanding advantages of the maximum entropy strategy in the continuous control of the manipulator through systematic comparative experiments.

However, the reward sparsity in goal-conditioned tasks of robotic arm control can cause traditional RL algorithms to fail. For example, for a typical robot grasping scene, the target object is an object randomly distributed in the workspace. In such a scenario, it is difficult to define the reward function because, if the distance between the end and the target is simply applied, the goal of avoiding obstacles cannot be expressed. At the same time, the premise of solving the accurate distance is to have clear modeling of the environment, obstacles, and targets. In this case, it is not meaningful to apply reinforcement learning. However, if a simple 0–1 variable is used as the reward function, most rounds cannot bring about state change, resulting in extremely low learning efficiency, which means the learning agent will not be able to receive any reward signal until eventually achieving the desired goal [43].

Many endeavors have been made to improve learning efficiency of reinforcement learning algorithms in the typical continuous control problem of motion planning of robotic arms and deployed a series of practical examples on real robotic arms [44,45,46,47], such as the deep deterministic policy gradient proposed by Lillicrap et al. [44] and the soft traveler critic proposed by Haarnoja et al. [45]. Outstanding achievements have been made in improving accuracy of state and environment evaluation. The applicability of these algorithms in the field of robot control has also been widely verified by real machines on a variety of complex tasks, such as posture arrival, handling, grasping, and solving magic squares [48].

In our unmanned chemical laboratory system, to address this sample inefficiency problem caused by sparse reward, we proposed a magnetic field-based reward shaping (MFRS) method for goal-conditional RL tasks with multiple targets and obstacles. Reward shaping [49,50,51,52] is an effective method to incorporate human domain knowledge into the process of policy learning and also maintains optimal policy invariance. Existing methods generally use a distance metric to calculate the shaping reward and may fail to carry sufficient information of learning tasks. As shown in Figure 6a, we utilize the properties of permanent magnets to establish the reward function in the field of the target and obstacles and concurrently learn a secondary potential function to convert the magnetic reward function into the form that satisfies the optimal policy invariance. With the nonlinear and anisotropic distribution of the magnetic field intensity, the MFRS method is capable of providing a more informative and explicit optimization landscape compared to the distance-based reward-shaping approaches, where S is a batch of states, A is a batch of actions, S’ is a batch of next states, R is the corresponding original rewards received from the environment, $F^{M}$ is the corresponding shaping rewards derived from DPBA based on magnetic rewards, and deep deterministic policy gradient (DDPG) [53] is a model-free actor–critic algorithm that can operate over continuous state and action spaces. Experiment results in simulated and real-world tasks both demonstrate the effective robot manipulation ability and improved sample efficiency. Figure 6b illustrates the success rate in each learning episode for various methods in simulation environments, where we evaluate MFRS in comparison to four baseline methods.

Control of Mobile Robots. Deep reinforcement learning (DRL) has great potential in complex decision-making and control problems, including robotic manipulation, legged locomotion, robotic navigation, multi-agent game, and other fields [55,56,57,58]. In the field of mobile robots, researchers try to use DRL to train robots to complete different tasks on the basis of lacking accurate physical models. Currently, the commonly used DRL-based training methods include end-to-end learning from scratch with a given task reward and imitation learning with some reference trajectory clips. Hwangbo et al. [24] successfully used the trust region policy optimization (TRPO) algorithm [59] to train a quadruped robot to complete different tasks, such as following high-level body velocity commands, running faster than ever before, and recovering from falling. Tsounis et al. [60] divided the motion of the quadruped robot into gait planning and gait control and used the proximal policy optimization (PPO) algorithm and model-based motion planning to train the quadruped robot to walk in the unstructured simulation environment. Fu et al. [61] used PPO algorithm to train the hexapod robot to move in an uneven plum blossom pile environment and used simplified kinematic and dynamic constraints to ensure rationality of the motion trajectory. The trained policy was verified in both simulation and real environments. Haarnoja et al. [62] proposed the soft actor critical (SAC) algorithm based on the maximum entropy reinforcement learning paradigm and trained an under-actuated 8-DOF quadruped robot to walk about 2 h end-to-end in the real world. Different from traditional reinforcement learning, imitation learning provides an imitation trajectory object, which makes the behavior of the agent more natural, and at the same time uses the prior trajectory information to accelerate the whole training process. Peng et al. [63] combined trajectory imitation with task objectives and used the paradigm of imitation learning to train legged robots to learn high dynamic behavior from a wide range of example motion clips, and the trained policy can respond to external disturbances robustly. Using the paradigm of GAN, generative adversarial imitation learning (GAIL) teaches agents to imitate target trajectory distributions. It solves the problem that traditional imitation learning can only imitate a single motion trajectory and suffers from accumulated trajectory errors. Peng et al. [64] proposed adversarial motion priors (AMP) algorithm based on the paradigm of GAIL, which enables agents to learn different behavior styles by using unstructured motion clips. However, the current DRL algorithm has low sample efficiency and is difficult to complete training tasks directly in the real world. In fact, it requires many trial-and-error processes in the simulation environment, which can be faster, safer, and more informative. How to transfer the trained policy from the simulation to reality to enhance the application value of the DRL is a key issue, known as Sim2Real transfer, which has also attracted the attention of many researchers [65].

Compared with wheeled robots, legged robots can pass through challenging environments by planning the landing points sequences and have broad application prospects [66,67], which makes control of legged robots more challenging. Hence, it is critical to solve the legged motion planning problem, and the effective control methods for legged robots are also of great importance for wheeled robots and robotic arms. For example, we propose a DRL-based multi-contact motion planning method for hexapod robots moving on randomly generated uneven plum blossom piles, as shown in Figure 7, where we design a policy network $\pi_{\theta }$ to map the current state $s_{t}$ of the hexapod robot to the action $a_{t}$; the proprioceptive information is represented by $\Phi_{t}$, $p_{target}$ is the center coordinate of the target area, and $M_{p}$ is the coordinates of all the plum-blossom piles. We feed the coordinates of all randomly distributed plum-blossom piles in the environment into the graph attention network (GAT) [94]. The output of GAT is concatenated with the remaining part of state $s_{t}$ and subsequently input into the multilayer perceptron (MLP), where FC is the fully connected layer and O is the output layer. According to $a_{t}$, the target footholds are found by the K-nearest neighbors (KNN) algorithm. The motion of the hexapod robot is formulated as a discrete-time finite Markov decision process (MDP) with a specific reward function. The designed DRL algorithm is used to generate the center of mass (COM) and feet trajectories of the hexapod robot. The inputs of the policy network include the robot’s exteroceptive measurements, the coordinate of the target area, and the environmental features extracted from the graph convolution neural network. In order to judge the transferability between two adjacent states, we propose a transition feasibility model based on the simplified single rigid body dynamic model and the trajectory optimization technique, and then the corresponding reward is determined. The trained policy is evaluated in different plum blossom pile environments, and both the simulation and experimental results on physical systems demonstrate the feasibility and efficiency of the DRL-based multi-contact motion planning method.

### 3.3. Computer Vision for Unmanned Chemical Systems

#### 3.3.1. Computer Vision for Robotic Manipulation

Real-time perception of the environment plays an important role in control of robotic manipulation. The effect of real-time perception depends on the performance of the sensors. Compared with mechanics pressure sensors [95], laser distance sensors [96], millimeter-wave radar [97], use of cameras-based vision solutions to achieve flexible planning of robotic arms [98] is undoubtedly the most appropriate in the chemical laboratory from the consideration of both accuracy and economy.

Robotic hand-eye systems in unmanned chemistry labs fall into two categories [99,100,101]: one is position-based vision servoing (PBVS) using two stages of vision-based positioning and path-planning for robotic arms; the other mainly adopts the ‘end-to-end’ idea and does not use the understandable but complicated algorithmic process of “visually determine the target and obstacle locations-determine whether the robot arm is impacted at each joint-solve the rational planning of the robot arm” to complete manipulation of the robot arm. The latter directly uses the image taken by the camera as the input variable for robot arm planning, which is also called image-based visual servoing (IBVS).

Actually, in real-world applications, position-based vision servoing often uses images to determine the pose of the target while utilizing position, and image-based vision servoing often includes algorithms such as collision detection based on position data. This mixture is independently referred to as hybrid-based vision servoing (HBVS) [102].

In position-based visual servoing, the scenario has very strong requirements for consistency between model and reality. To address this, Sharifi et al. proposed a new iterative adaptive extended Kalman filter algorithm to estimate the position and pose of an object in real time to improve positional estimation accuracy in the presence of certain model errors and noise [103]. Wang et al. proposed a dynamic model based on data, which connects information utilization between distance judgment and manipulator planning and achieved a more stable control effect [104]. For relatively fixed scenes, such as unmanned chemical laboratories and industrial assembly lines, adding some target marks can undoubtedly greatly enhance efficiency of object recognition. A typical target mark usually consists of an array of easily recognizable graphic elements, such as squares, circles, and lines.

In image-based vision servoing, visual information is used to obtain the characteristics of the observed object in image space, and the error is formed by comparing it with the desired image characteristics, and a control law is set based on this error to control the robot or camera motion. Its strategy of relying on the current image to complete control also causes its control information to be limited to the “current field of view”, which may result in control failure for large motion scenes [105].

The primary requirement of traditional graphical IBVS is to complete extraction of picture features to complete the comparison between the target state of the robotic arm and the current rotation state. The commonly used machine vision feature extraction algorithms in this process are SIFT [106], Daisy [107], SURF [108], ORB [109], etc. Many current works use convolutional neural networks to complete the entire process from feature extraction to control scheme generation, eliminating the errors arising in feature extraction and comparison. Saxena et al. train Flow Net using synthetic images to predict the current pose of the camera and complete visual servo operation by comparing it with the target pose [110]. Bateux et al. train the VGG network to directly predict the relative positional error between two images [111]. Neuberger et al. first use the pose estimation network to complete the initial pose tracking and then use a vision feature extraction algorithm to complete fine-tuning, taking full advantage of the high accuracy of IBVS steady-state [112].

#### 3.3.2. Object Detection

In unmanned chemical experimental systems, visual recognition involves target identification and detection, and detection is the prerequisite for robotic manipulation tasks. However, due to the complex and diverse environments of chemical laboratories, a variety of chemical instruments and equipment may become targets for recognition by the detection system. The distance and location between each target and the robot are different and the size of the target in the image varies greatly. In addition, given that the images taken in the laboratory scene are affected by the brightness of light and darkness and mutual occlusion of each target, it is greatly challenging in application of computer-vision-based object detection.

At present, there are two types of object detection methods: traditional image processing and deep learning methods. Researchers have been focusing on deep learning and convolutional neural networks (CNN) since the AlexNet [113] algorithm achieved first place in the ImageNet competition in 2012. At present, the mainstream deep-learning-based target recognition detection methods contain three categories: first-order target recognition detection, second-order target recognition detection, and anchor-free target recognition detection. One-stage object detection is represented by the YOLO and SSD series algorithms. The image input to the convolutional neural network will directly obtain a rectangular box identifying the detected target, the target category, and the probability judged by the algorithm. Two-stage object detection (for example, R-CNN [114]) is more complex, more expensive to train, and slower to recognize but with higher recognition accuracy compared with one-stage object detection. Anchor-free object detection directly detects the target without a preset anchor frame for detection and currently can be divided into key-point-based detection methods and center-point-based detection methods, where CornerNet [115], CenterNet [116], FCOS [117], etc., achieve better results.

There have been many attempts at target identification and detection in unmanned chemical environments. Sagi Eppel et al. evaluated the correspondence of real liquid surface contours to achieve recognition of liquid surfaces and liquid levels in various transparent containers with a miss rate of less than 1% for gas–liquid surfaces [118], and, in 2016, Sagi Eppel et al. proposed a graph-cutting algorithm capable of being used to identify the boundaries of materials in transparent containers with high accuracy for identification of the boundaries of liquid, solid, granular, and powder materials in various glass containers in chemical laboratories [119]. In 2018, Solana-Altabella et al. developed two different computer-vision-based analytical chemistry methods to quantify iron in commercial pharmaceutical formulations with less than 2% error [120]. In 2020, Sagi Eppel et al. proposed a machine learning approach and provided a new dataset, Vector-LabPics, for identifying materials and containers in chemical laboratory containers [121]. In the same year, Tara Zepel et al. developed a generalizable and inexpensive computer-vision-based system for liquid-level monitoring and control [122]. In 2021, Kosenkov proposed a new technique for automated titration experiments using computer vision [123].

Although the current computer-vision-based recognition and detection technology has helped to improve the efficiency of chemical experiments, most of the existing unmanned chemical experiment systems are designed with multiple dedicated chemical automation devices, which results in high cost and poor versatility of unmanned chemical experiment systems. It is still a great challenge to develop efficient and practical computer-vision-based object recognition and detection systems for automatic unmanned chemical laboratory systems.

### 3.4. Intelligent Scheduling for Unmanned Systems

Design of an automation system framework possessing the function that supports users regarding remote operation (e.g., plan experimental workflow, execute and monitor experiments, etc.) is another important aspect to improve the degree of automation. Meanwhile, to make the system allocate experimental resources to several tasks reasonably, improve equipment utilization efficiency, and reduce the total time of tasks, the system requires intelligent scheduling to optimize the process of the whole system.

Scheduling is defined as scheduling of resources, including machines and materials used by several tasks, in time so that one or more objective functions of the whole system can be optimized. As one of the key problems in the field of intelligent manufacturing, scheduling problem has been widely used in enterprise management, production management, and scientific research. In view of the concept of scheduling, it has the features of objective diversity, environmental uncertainty, and the complexity of solving the whole problem. For the diversity of objectives, the general aim of production scheduling is based on the combination of scheduling plan and evaluation index. The diversity of scheduling objectives is caused by different manufacturing environments. At the same time, the scheduling index requires to consider non-unique indicators, including the lowest cost and the highest utilization rate of the equipment. Consequently, the design scheduling should pay attention to avoid conflict between indicators. For uncertainty, there are many random uncertainties and other factors in production scheduling, e.g., equipment failure and communication failure. The real environment may not be consistent with the plan, which requires dynamic scheduling to fully adapt to working environment changes. For complexity, objective diversity and uncertainty cause the complexity of solving the production scheduling problem.

During development of scheduling, several solutions have emerged, e.g., job-shop problem based on genetic algorithm, flow-shop scheduling problem adaptive simulated annealing algorithm, genetic algorithms for changing structural space, and neural network for solving dynamic scheduling problems. Generally, production scheduling can be divided into two parts: the first is classical scheduling theory, which is a scheduling method based on operational research theory. The second is intelligent scheduling, which is based on artificial intelligence technology. Classic scheduling is a mathematical model based on operations research to solve the optimization model with constraints. Due to the real environment being dynamic, it is hard for classical scheduling models to adapt to the complexity of scheduling problems. However, based on artificial intelligence technology, assembly of intelligent equipment that integrates self-perception, self-learning, data analysis, decision generation, and production adjustment can be well suited to production practice.

For chemistry automation, it is necessary to coordinate various types of instruments to complete the experiment, avoid resource conflicts, and reduce the execution time of the experiment. To solve the lab tasks scheduling problem, Cabrera et al. used a simulated annealing algorithm to solve dynamic scheduling of experimental tasks [124]. Alexander et al. used ant colony algorithm to solve task scheduling in the mineral laboratory and compared it with the exact method to verify its effectiveness [125]. Li et al. described sample detection in a medical laboratory as a parallel batch online scheduling problem and solved it with the objective of minimizing maximum weighted flow time [126]. Takeshi et al. described the lab automation scheduling problem with time constraints between operations as a mixed integer programming problem and proposed a scheduling method based on the branch and bound algorithm [127]. Shan et al. used the improved genetic algorithm to solve the biological laboratory scheduling problem. Wu et al. defined the task assignment problem of metrological detection in an unmanned metrological laboratory as an open shop scheduling problem and proposed a discrete gray wolf optimization algorithm based on fitness and distance evaluation criteria to solve it.

We also designed a framework of automated chemical synthesis system, which implements the functions of experiment scheme formulation, experiment flow monitoring, experiment equipment management, and user interaction interface. By unifying the software and hardware interfaces and communication protocols, integration of experimental equipment is more efficient and can satisfy the diverse experimental requirements of users. In the design of the experimental scheme, the system can directly implement the required experimental process in the interface of net. The web page of the unmanned chemical system is shown in Figure 8.

To improve the experimental efficiency of the system and achieve the best match between resources and tasks, it is necessary for the computer to comprehensively calculate the experimental equipment, materials, and requirements. Finally, the system obtains the optimal scheduling scheme. In the scheduling algorithm, it is necessary to consider use of materials and tubes, reaction conditions, and time required by different experimental processes and coordination of the mechanical arm, liquid transfer station, product detection station, and other devices to reduce the idle time of the equipment. In intelligent scheduling, optimization of scheduling can be regarded as a reinforcement learning problem, and the reinforcement learning process of searching the best policy is used as the optimization process of searching the optimal solution. Zhang et al. proposed a reinforcement learning method to minimize average weighted delay in irrelevant parallel machine scheduling [128]. Gyoung et al. proposed genetic reinforcement learning to solve the scheduling problem as an RL problem [129]. Jamal et al. considered random job arrival and machine failure and adopted reinforcement learning of Q-factor algorithm to improve the dynamic job shop scheduling problem [130].

### 3.5. Augmentation of Automatic Chemical Systems with Computation-Based Screening

Despite that the automatic chemical synthesis system can liberate experimenters and make the accessible data in the order of thousands, the high-dimensional parameter spaces make chemical research with high-throughput experiments cost-prohibitive. For instance, optimization of the maximum reaction yield for a target product always needs to explore hundreds or thousands of possible reaction condition combinations (e.g., catalysts, ligands, substrates, additives, etc.). This contributes to high time- and material costs for random high-throughput experiments with automated synthesis. Thus, it is highly desirable to introduce other tools to lower the scope and parameters for automated experiments. With advances in computer science and computational methods and software, a diverse array of high-throughput or automated computational toolkits are now available. We believe integration of the automatic reaction machine with computational-based virtual screening will not only augment the intelligence of automatic chemical systems but also makes the exploration process more time- and resource-efficient. Recently, Jiang and co-workers developed a comprehensive artificial intelligence chemistry laboratory. The platform incorporates a machine reading module, a mobile robot module, and a computational brain module and can target different chemical tasks [131]. An investigation of dye-sensitized photocatalytic H_2_ productions with an average of deviation bars of 5.5% for H_2_ production rate and 8.3% for RhB degradation efficiency demonstrates the high accuracy and repeatability of the platform. Here, we will take chemical synthesis as an example to facilitate augmentation of automatic chemical systems with computational-based simulation from three aspects, including quantum-mechanical-based automated reaction design, purely data-driven reactivity prediction models, and retrosynthetic route planning.

In past decades, quantum-mechanical (QM) calculations have achieved great success both in discovery of new mechanisms and new reactions [132,133,134]. Although the data-driven approach tends to be more efficient in reactivity prediction of the chemical reactions, the QM-based new reaction design remains highly indispensable because it is not only an accurate approach for modeling chemical reactivity but also can provide rich mechanistic insight for reactions. More importantly, several automated reaction pathways searching methods have been developed, such as MD/CD, Q2MM, LASP, etc. [135,136,137,138,139]. These automated computational workflows offer powerful and low-labor-intensive tools for new reaction and catalyst design due to their low or no human intervention. Further, through combination with machine learning techniques, accurate and efficient automated reactivity prediction tools would be achieved to realize rapid prediction of chemical reaction mechanisms as well as high-throughput virtual screening [140]. Then, candidate reactions would be discovered and recommended for automatic chemical systems and the latter further carry out experimental verification accordingly. Thus, such a paradigm will reduce the number of trial-and-error experiments required for new reaction discovery.

In addition to QM-based reactivity prediction, fast strategies based on purely data-driven ML methods would be highly useful to improve the material efficiency of automated chemical reaction systems. Recently, machine learning (ML) methodologies were demonstrated to be useful in prediction of reactivities, regio-, and stereoselectivity [31,32,33,34,35]. For example, Doyle et al. reported the performance of the random forest model on the prediction of the reaction yield of a Buchwald–Hartwig amination reaction [31]. A dataset of 4608 reactions generated through high-throughput experiments (HTE) on four-parameter combinations (including twenty-three isoxazole additives, fifteen aryl and heteroaryl halides, four palladium catalysts, and three bases) was used for modeling. The final model predicted the yield with root mean square error (RMSE) of 11.3% and R^2^ of 0.83 on a series of out-of-sample additives. Coupling this approach with the experiments might facilitate one to explore the reaction reactivities or optimal conditions of new substrates. Machine-learning-guided workflow for improving the efficiency of automated experimentations generally consists of four steps: (1) data collection from automated experiments or literature reports; (2) feature selection and building reactivity/selectivity prediction model; (3) virtual screening on a large reaction space; (4) candidate reaction recommendation; (5) verification through automated synthesis; (6) feedback and iterative to retrain the model for performance improvement; (7) discovery of new reactions. In comparison with QM-based screening, the purely data-driven ML approach requires the lowest computational cost, and we, therefore, believe that introduction of an ML-assisted reaction discovery approach to the automated chemical synthesis system will significantly improve efficiency.

The aforementioned reactivity prediction strategies are mainly focused on development of new reactions or process optimization. Another important aspect is how to efficiently construct the target functional molecules (e.g., natural products, pharmaceuticals, agro-, and fine chemicals, etc.) with the automated chemical synthesis machine. Computer-assisted retrosynthesis is a valuable tool for planning the synthetic route of the desired structures, and ML has achieved great success in this field [68,141,142,143]. For instance, Waller and Segler et al. [68] combined deep neural network policy learned from the published knowledge of chemistry, Monte Carlo tree search (MCTS), to guide the search and filter the network to preselect the most possible reverse synthesis route [68]. The approach was modeled on 12.4 million reactions from the Reaxy database, and the results showed that the model was 30 times faster than the known computer-aided search method based on rule extraction and manual-designed heuristic. Compared with the 87.5% accuracy of the benchmark forecast, the accuracy of this strategy is 95%. With a designed synthetic route, automated chemical systems can execute the synthesis task to obtain the target molecules. For example, based on the continuous flow reaction system, Jamison and Jensen et al. developed an open-source kit, ASKCOS, for computer-aided synthesis route design (CASP) [141], which was used to plan the molecular synthesis route, design the operating conditions according to the prior reaction efficiency information (e.g., the residence time of each part of the continuous flow reaction, the amount and concentration of reactants and products). Then, that synthetic information was refined into a chemical “formula”, which was then sent to the robot platform to automatically assemble hardware and conduct the synthetic operation to build molecules.

## 4. Conclusions

This paper first reviewed unmanned chemical synthesis systems and provided a basic introduction to machine learning. Second, we expounded on the problems existing in the current unmanned chemistry systems. Finally, the prospect for strengthening the connection between reaction pathway exploration and the existing automatic reaction platform and solutions for improving autonomation through combination with machine learning algorithms, such as information extraction, control of robots, computer vision, and intelligent scheduling, were proposed. In addition to the aforementioned aspects, there is great power and potential in application of machine learning models for chemistry, such as property predictions, material design, molecular simulations, and chemistry grammar extraction [144,145,146,147,148]. In the future, merging of data science and machine learning algorithms with automation techniques will revolutionize unmanned chemical synthesis systems toward higher intelligence.

## Figures and Tables

**Figure 1 molecules-28-02232-f001:**
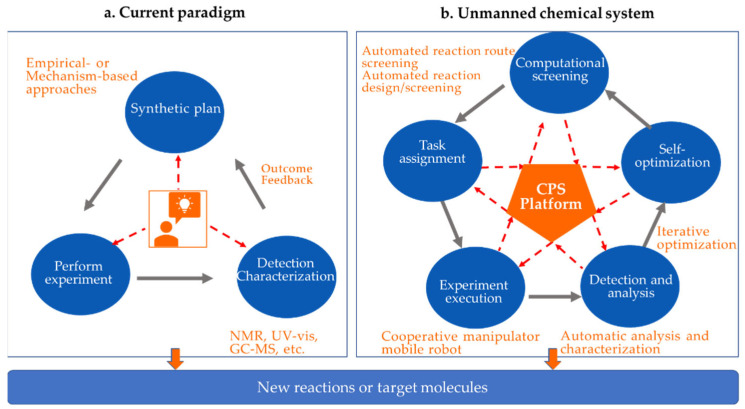
Schematic comparison of chemical synthesis paradigms. (**a**) The current paradigm; (**b**) reaction discovery or molecule synthesis with unmanned chemical systems integrated by the CPS (cyber-physical system) platform.

**Figure 2 molecules-28-02232-f002:**
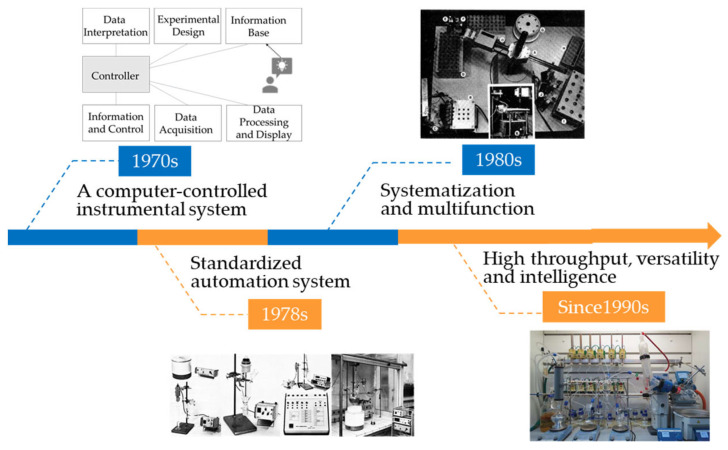
Development of automated chemical synthesis platform. Picture in 1970s adapted with permission from Ref. [1]. Copyright 1971, American Chemical Society. Pictures in 1978s adapted with permission from Ref. [2]. Copyright 1978, American Chemical Society. Pictures in 1980s adapted with permission from Ref. [3]. Copyright 1984, American Chemical Society. Picture in since 1990s adapted with permission from Ref. [4]. Copyright 2018, The American Association for the Advancement of Science.

**Figure 3 molecules-28-02232-f003:**
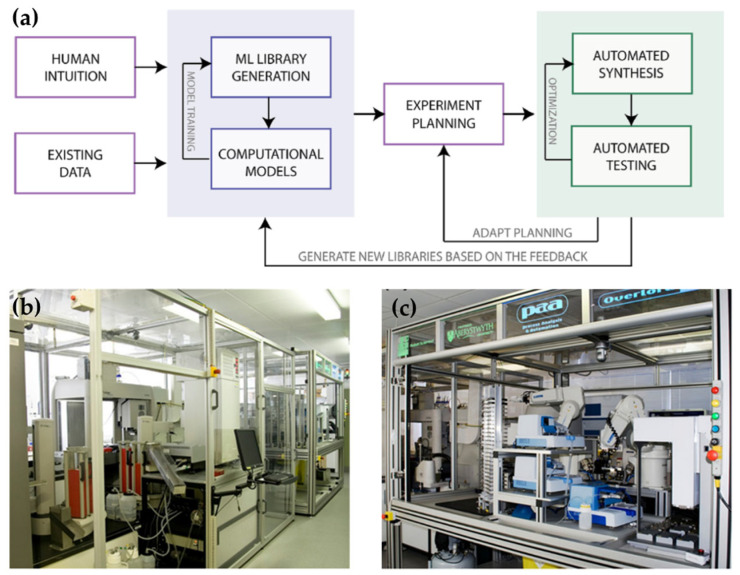
Unmanned chemical system. (**a**) Schematic illustration of an autonomous discovery system; (**b**) Adam’s laboratory robotic system; (**c**) Eve’s laboratory robotic system. Figure 3a is reprinded with permission from Ref. [25]. Copyright 2019, American Chemical Society. Figure 3b,c is adapted from Ref. [27] and used under CC BY 2.0.

**Figure 4 molecules-28-02232-f004:**
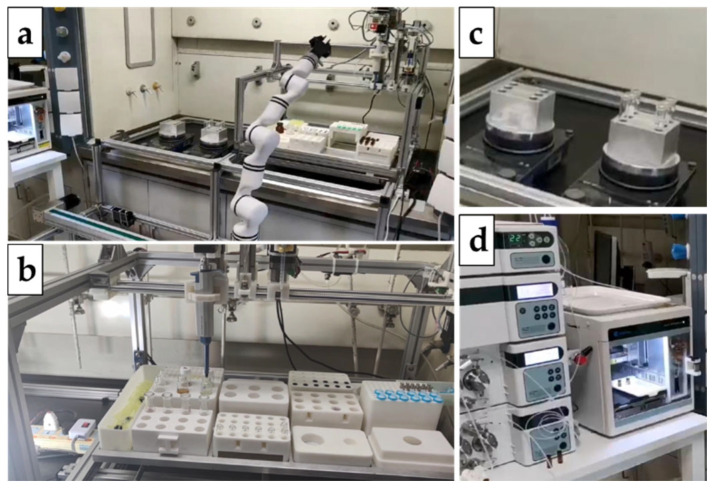
Illustration of an unmanned chemical system, where (**a**) is the robotic arm, (**b**) is the pipetting station, (**c**) is the reacting region, (**d**) is the characterization module (e.g., HPLC).

**Figure 5 molecules-28-02232-f005:**
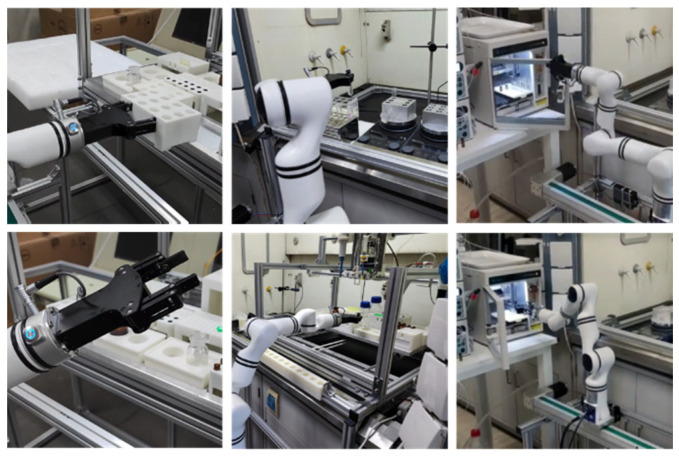
The robotic arm in our unmanned chemical laboratory carries out elaborate operations, such as grasping and transferring reaction tubes or sample vials and manipulating other instruments.

**Figure 6 molecules-28-02232-f006:**
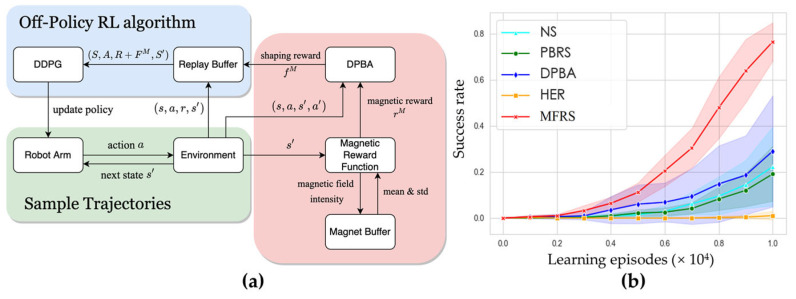
A reinforcement learning method using magnetic-field-based reward shaping (MFRS) for robotic arm control. (**a**) Overview of MFRS for goal-conditioned RL tasks with multiple targets and obstacles. (**b**) Success rate per learning episode of MFRS with comparison to baselines trained in simulation environments. NS: training the policy with the original reward given by the environment without shaping. PBRS [49]: potential-based reward shaping, which adds a calculated shaping reward to the original reward, where the potential function is built on Euclidean distance. DPBA [52]: dynamic potential-based advice, which can transform any given reward function into the form of potential-based reward shaping to ensure the optimal policy invariance property. HER [54]: hindsight experience replay, which relabels the desired goals in the replay buffer with the achieved goals in the same trajectories. We adopt the default “final” strategy in HER that chooses the additional goals corresponding to the final state of the environment.

**Figure 7 molecules-28-02232-f007:**
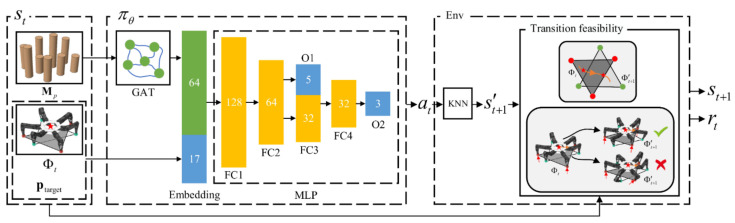
Overview of the DRL-based multi-contact motion planning method for hexapod robots.

**Figure 8 molecules-28-02232-f008:**
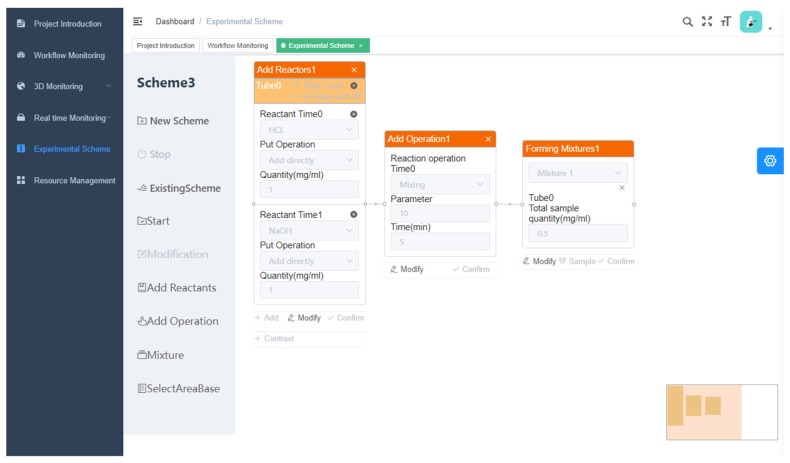
Illustration of the web page of the unmanned chemical system. The leftmost menu is the system administration section. The central section of the diagram shows the core content of the experimental design of an unmanned chemical system. First, we can drag ‘Add Reactants’, ‘Add Operation’, and ‘Mixture’ from the left menu to the empty configuration area on the right and fill in the required parameters. Then, we can click ‘Start’ to start the unmanned chemical system and the chemical experiment.

## Data Availability

Not applicable.

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
