# Peer review of "Machine Learning in Unmanned Systems for Chemical Synthesis"

_molecules, 2023, doi:10.3390/molecules28052232_

Round 1

Reviewer 1 Report

Here is my opinion on the manuscript molecules-2192609 entitled "Machine Learning in Unmanned Systems for Chemical Synthesis" by Guoqiang Wang et al.

The manuscript provides a useful overview of the knowledge and results in the field of optimisation and development of systems for guidance/control of chemical synthesis. In general, it should be noted that the review lacks critical reviews and opinions/recommendations on how a particular segment of the chemical synthesis guidance or prediction system can be improved in the future. Furthermore, the authors and editors of the journal have to take into account that many parts of the text and terms (names, concepts) from the technical sciences (robotics, automation) are not self-explanatory for scientists from the field of chemistry - who are the most numerous readers of Molecules.
Although the literature review is quite technical in many parts (from a chemical point of view) - without the explanation of processes and the chemical background of successful systems/solutions (which should be remedied as much as possible) - this manuscript deserves to be published in Molecules after the necessary corrections.

Comments:
1) The impression one gets from reading this reviewed paper is that the systems/solutions/tools developed are perfect and it looks like they solve the problem 100% correctly. In reality, we know that every solution works with some error (or is less than 100% accurate). Therefore, it is necessary to point out this aspect - accuracy/error - when reviewing the published solutions and discussing possible improvements.

2) For each result/solution from the literature described in the review (e.g. those listed in Table 2), a random accuracy can be calculated or estimated in addition to the accuracy obtained and described. This is illustrated, for example, by the example of retrosynthetic predictions for binary classification models in the paper https://doi.org/10.3390/molecules25102357. We believe that this work is worth mentioning in the review. Furthermore, if it is possible to estimate the random accuracy for individual models (which is calculated for binary classification problems according to the formula given on page 7 of the aforementioned paper - with reference to references 25 and 26 from this paper), it is also worth mentioning and is given with the actual model accuracy - which is also simply called accuracy in classification ML modelling.

3) I noticed that in the paper the first (and only) table is referred to as Table 2 (see line 124). This needs to be corrected in the table and in the text - Table 2 must be re-named in Table 1.

4) To make the content more comprehensible, it is necessary to note and describe the abbreviations and terms marked in the figures in the text. As an example of inadequacies, I will only describe Figure 7 - i.e. many abbreviations are not explained and are not self-explanatory (GAT, FC1, FC2, ..., O2, St, etc.).

Reviewer 2 Report

The manuscript titled "Machine Learning in Unmanned Systems for Chemical Synthesis" by Wang et al. is a comprehensive review of recent advancements and publications in this field. The focus is on the use and implementation of machine learning for efficient chemical reaction monitoring and synthesis. The review covers a wide range of publications and attempts to fill any gaps. As a reviewer, I have read the entire manuscript and have not identified any major revisions that are needed. However, there are several spelling and grammar errors present throughout. I would recommend reducing the word count to make the review more concise and to eliminate any unnecessary information. For example, from lines 314-316, it is not necessary to include a translation in English, as it does not add to the content of the manuscript.

Author Response

Response 1: Thank you very much for your positive comment. We have carefully revised our manuscript according to your constructive suggestions.